# Fully Automated and Robust Cable Tension Estimation of Wireless Sensor Networks System

**DOI:** 10.3390/s21217229

**Published:** 2021-10-30

**Authors:** Min Zhang, Huating He, Gengying Li, Haiyang Wang

**Affiliations:** College of Water Conservancy and Civil Engineering, South China Agricultural University, Guangzhou 510642, China; ammindy@scau.edu.cn (M.Z.); hehuating@stu.scau.edu.cn (H.H.); ligengying@scau.edu.cn (G.L.)

**Keywords:** cable tension estimation, fully automated, wireless sensor networks

## Abstract

Accurate estimation of cable tension is crucial for the structural health monitoring of cable-supported structures. Identifying the cable’s force from its vibration data is probably the most widely adopted method of cable tension estimation. According to string theory, the accuracy of estimated cable tension is highly related to identified modal parameters including natural frequencies and frequency order. To alleviate the factors that impact the accuracy of modal parameters when using the peak-picking method in wireless sensor networks, a fully automated and robust identifying method is proposed in this paper. This novel method was implemented on the Xnode wireless sensor system and validated with the data obtained from Jindo Bridge. The experiment results indicate that, through this method, the wireless sensor is able to distinguish the cognizable power spectrum, extract the peaks, eliminate false frequencies and determine frequency orders automatically to estimate cable tension force without any manual intervention or preprocessing. Meanwhile, the results of natural frequencies, corresponding orders and cable tension force obtained from the Xnode system show excellent agreement with the results obtained using the Matlab program method. This demonstrates the effectiveness and reliability of the Xnode estimation system. Furthermore, this method is also appropriate for other high-performance wireless sensor network systems to realize self-identification of cable in long-term monitoring.

## 1. Introduction

Cable-supported systems are widely used in large span space structures, such as air-supported roofs, cable-stayed bridges, suspension bridges, cable domes, and so on [1]. As main load bearing components, cables play important roles in these structures and their failure may result in the structure’s accidental collapse. Furthermore, the condition of cable-stayed structures can be assessed through their cable tension during operation [2,3]. There are many methods to measure cable tension force, such as using load cells, hydraulic jacks [4], FBG sensors [5] or electromagnetic sensors. However, those conventional sensors become unfeasible and impracticable when applied in long-term monitoring. For example, the method using load cell is uneconomical since the service life of a load cell is generally much shorter than that of a structure. FBG sensors are sensitive to environment or operating technology, thus have lower accuracy. Some of these sensors must be installed during construction and are difficult to maintain, something that is not amenable to existing structures [6,7].

The vibration-based method is commonly employed to estimate cable tension as it is fast, economical and easy to operate compared with the above mentioned methods [8]. This dynamical method is capable of calculating cable tension force directly using only the cable’s acceleration responses through the relationship between the cable’s natural frequencies and the cable’s tension [9,10,11]. Zui et al. proposed a simple formula to estimate cable tension by using measured natural frequencies in low-order modes and calibrating the results with finite element analytical results [12]. Ren et al. developed an empirical formula to obtain cable tension based only on the cable fundamental frequency and verified it with experimental results [13].

With the rapid development of internet of things (IoT)-enabled wireless sensor networks (WSN) [14,15], smart wireless sensors, with the capability to process data locally and transmit information through wireless communication, have become increasingly attractive and widely utilized in many fields [16,17,18], including structure health monitoring, mechanism control and big data analytics. IoT-WSN technology is particularly applicable for cable tension monitoring [19,20,21,22], especially for those large scale cable-supported structures, with a large number of cables, which are a great challenge to wired sensor monitoring systems or traditional centralized process technologies [23]. Cable tension estimation via wireless sensor technology has been the subject of major research in recent years. For example, considering the effects of the sag-to-span ratio and bending stiffness, Cho et al. [24] proposed an automated and low-cost wireless sensor system for the continuous monitoring of cable tension based on Zui’s approach. The peak-picking method has been used with a Fourier spectrum to identify modal frequencies, which were then utilized in cable tension estimation. Sim et al. [25] proposed a hybrid wireless sensor network based on the Imote2 platform to detect cable tension by using simple string theory for full scale cable-stayed bridges. To extract highly reliable frequencies from the power spectrum, a threshold value was set to search the highest points by comparing local data near natural frequencies. This method had been implemented on Jindo Bridge, located in Korea, and its effectiveness had been verified successfully. A smartphone-based portable cable tension testing method has been proposed by Zhao in which the cable force can be calculated by the estimated model frequency of cables through a software named Orion CC [26,27].

In order to make full use of wireless sensor networks in cable tension estimation (CTE) systems and ensure the system satisfies the requirements of many different types of cable-supported structures, it is essential to develop a fully automated and robust cable tension estimation system. Previous research has achieved successful results in cable tension force monitoring, however, there are still some problems to be solved, which mainly include: (1) Preprocessing or manual interventions required during testing, such as pre-calculated reference modal parameters or picking peaks manually, are usually inescapable [25,26,27]. This is a big challenge for large-scale cable structures with a dense array of sensors. (2) The peak-picking method is commonly and conveniently used in extracting frequencies, especially for clearly separated power spectra, but in practice the power spectra of acceleration responses from ambient excitation usually have undesirable peaks or miss the inner peaks. Threshold values are helpful in picking reliable peaks, but fail to eliminate false frequencies. Meanwhile, the peak-picking method is particularly related to the vibration mode, and so may fail to identify the modal parameters for weak vibrations. (3) The current systems are mainly used for special structures or applications, which are not universal, so the properties and vibration characteristics differ widely. This limits the development and applications of an estimation system. 

In this study, a robust and fully automated cable tension estimation system is proposed. Firstly, the main factors affecting the accuracy of cable tension estimation are discussed with regard to instances where the peak-picking method and simple string cable model are used in cable tension estimation via wireless sensor networks. Furthermore, a CTE strategy is developed to solve the potential problems resulting from incorrectly identified frequencies and corresponding mode orders. Finally, this method is implemented with the Xnode wireless sensor system, and the data collected from Jindo Bridge employed to verify the feasibility of the system.

## 2. Theoretical Background of Cable Tension Estimation via Cable Vibration

The vibration-based method is one of the most convenient and widely accepted cable tension estimation methods, because it requires only an accelerometer to measure the cable vibration subjected to environmental loading. For cable-stayed structures, the cable can be simulated as an inclined string with two fixed ends. In this paper, the flat taut string theory is utilized to estimate cable tension via the vibration frequencies of the cable. Based on the assumption that the cable’s bending stiffness is considered and the sag-extensibility is neglected, the dynamic differential equation of the stayed-cable is given as follows:(1)$$m\frac{\partial^{2}v(x,t)}{\partial t^{2}}+EI\frac{\partial^{4}v(x,t)}{\partial x^{4}}-T\frac{\partial^{2}v(x,t)}{\partial x^{2}}=0$$
where m is the cable’s mass per unit length, v is the cable’s deflection, EI is the flexural stiffness of the cable, and T is the cable tension. Based on the tightening string model, the relation of the cable’s natural frequencies and the cable tension can be expressed by:(2)$${\left(\frac{f_{n}}{n}\right)}^{2}=(\frac{1}{4mL^{2}})T+(\frac{n^{2}\pi^{2}}{4mL^{4}})EI=a+bn^{2}$$
where $f_{n}$ is the natural frequency, n is the order of the natural frequency, L is the length of cable, and a and b are the linear regression results of ${\left(f_{n}/n\right)}^{2}$ and $n^{2}$. Using Equation (2), the cable tension force can be estimated as:(3)$$T=4\times m\times L^{2}\times a$$

## 3. Key Influence Factors of Cable Tension Estimation Accuracy

According to the principle mentioned above, it is obvious that the accuracy of cable tension is particularly related with the modal parameters, mass (*m*) and length of the cable (*L*). The inaccuracy of those parameters may result in a large error in the estimation of cable tension, something that will be discussed as follows. 

In practice, it is a big challenge for a wireless sensor to extract the cable’s natural frequencies autonomously and accurately from the power spectrum of the cable’s vibration through the peak-picking method. The commonly encountered problems are listed as follows:(1)Identifying useful vibration data.

In real testing, the cable vibration may not be strong enough to show obvious frequency poles in power spectrum. In this case, there may be no frequency peaks to be found or undesirable peaks may be eventually picked through the regular peak-picking method [25], as shown in Figure 1. Consequently, the obtained cable tension would not be correct. Figure 1a shows a cable vibration that is too weak to exhibit the obvious peaks. Figure 1b shows a power spectral density (PSD) with many false peaks found through the regular peak-picking method. In this instance, the wireless sensor should have the capability to identify whether the power spectrum of acceleration response would be useful and recognizable.

(2)Extracting a cable’s natural frequency from vibration data automatically and robustly.

For each cable, the auto-power spectrum density is generally adopted to extract structural natural frequencies by using its own vibration response. However, the acceleration signal may be influenced by many uncertain environmental factors [28,29], which may lead to some false peaks in the auto-power spectrum. The aforementioned theory indicates that the frequencies used in Equation (2) are all with correct mode orders. If the identified frequencies or the mode orders are inaccurate, then the cable tension force would not be correct. Thus, the automatic extraction of the true frequencies of the cable is also a problem to be solved for wireless sensor networks. 

(3)Identify the frequency order automatically.

Generally, the prominent peaks of the power spectrum density may not start from the first mode [25] or may exist in a discontinuous order. The example in Figure 2 shows that one modal frequency (marked with a black spot) is missed, meaning that the order of obtained potential frequencies will be distorted. In such a case, automatically identifying the mode order is also challenging for a smart sensor.

## 4. Automated and Robust Cable Tension Estimation Method

According to the theory mentioned in part two, assuming that unit mass and effective length of cable are constants, the cable tension estimation is focused on identifying frequencies and their orders automatically in WSN. Considering the aforementioned problems, an improved strategy based on vibration method is developed for a wireless sensor to realize fully automated and robust cable tension estimation. As shown in Figure 3, the whole method includes three steps: (1) automatically calculating PSD from acceleration data; (2) fully automated and robust peak-picking analysis to obtain true frequencies; (3) identification of the mode order and computation of cable tension force according to the linear regression result. 

The details of each step will be discussed as follows. 

### 4.1. Power Spectral Density Estimation

The PSD of cable acceleration vibrations is estimated using the Welch periodogram method, the algorithm was implemented in the Xnode wireless sensor platform and illustrated in Figure 4.

Firstly, the number of the segment (*k*) utilized to allocate memory for all acceleration data is automatically calculated based on their overlap and data length. Then the acceleration data are divided into *k* segments and saved temporarily as *acc*_1_, *acc*_2_, *acc_k_* respectively. 

Secondly, a fast Fourier transformation (FFT) is conducted on each segmented response with a Hanning window to obtain auto-power spectrum (APS) estimation of the acceleration response. Only half of the complex value results are saved because of symmetry of data so that a long-term history record segment can be processed by using a modest amount of memory.

Subsequently, the averaged power spectral density can be obtained by dividing absolute value of summary of $APS_{i}$ by k, with the results saved locally for the next process of the peak-picking algorithm. 

Finally, the memory of whole time domain segments is released to save memory. 

### 4.2. Automatic and Robust Peak-Picking Method 

Basically, the power spectrum of vibration data contains many peak responses which correspond to a cable’s natural frequencies, so that a simple peak-picking method with threshold value is used to extract the frequencies [25]. 

(1)Updating threshold value automatically for weak vibration.

The threshold value ($\varepsilon_{u}$) is initially set to be the sum of mean(psd)+2×std(psd) for peak picking with 95% confidence [25]. However, in practice, even though the power spectrum is clearly separate and cognizable, it is difficult to capture enough peaks with the initial $\varepsilon_{u}$ all the time, especially for some weak vibration, (e.g., 70% of the data from Jindo Bridge are in this situation).

In this case, the threshold value will be reduced automatically during the searching process until enough peaks are obtained or until the threshold value reaches the lower limit $\varepsilon_{l}$. If there are still not enough peaks, the cable tension task will be returned as zero. Figure 5 shows the procedure of picking peaks by adjusting threshold value. The number of identified peaks was increased from two to seven by updating the threshold value and, thus, the estimated cable tension became much more reliable.

(2)Exclude the false frequencies.

Figure 6 shows the power spectral density of one cable in Jindo Bridge. It exhibits several peaks above 8 Hz including the third peak marked by a red spot, which is a false frequency. Therefore, in practical vibration testing, it is quite possible to get undesirable peaks that are bigger than the threshold value. Meanwhile, for some useless vibration data, there may still be many peaks by using initial threshold value, such as in the example shown in Figure 1b. 

In WSN system, the true mode frequencies should be distinguished automatically for cable tension estimation. The detailed procedure is described as follows, with a flow chart in Figure 7.

During the implementation of the peak-picking process, the position number (order in frequency region) of each qualified peak is temporarily saved as $\begin{array}{ccccc} N_{-}ID_{1}, & N_{-}ID_{2}, & \ldots & N_{-}ID_{i}, & N_{-}ID_{n} \end{array}$. In principle, the difference value ($\Delta N_{-}ID_{i}=\Delta N_{-}ID_{i+1}-\Delta N_{-}ID_{i}$) between any two adjacent mode frequencies should be almost the same, which can be adopted to eliminate false frequencies. The most frequent value α in array of $\Delta N_{-}ID_{i}$ is at first acquired automatically. 

For the non-vibration data that still has many disordered peaks (in Figure 1b), the differences values ($\Delta N_{-}ID_{i}$) are generally randomly distributed, therefore the number of $\Delta N_{-}ID_{i}$ which approximates to α (in this paper $abs(\Delta N_{-}ID_{i}-\alpha )\leq 1$) would be less than half, through which the data could be determined as invalid data. Hence, the found peak is set as empty and cable tension returned as zero, CTE application is finished temporarily.

Any two adjacent $N_{-}ID_{i}$ and $N_{-}ID_{i+1}$ would be interpreted as true frequencies as long as the quotient of $\Delta N_{-}ID_{i}$ and α are much close to an integer. The $\left[\begin{array}{cc} l_{1} & l_{2} \end{array}\right]$ ($\left[\begin{array}{cc} 0.1 & 0.9 \end{array}\right]$ in the paper) are two threshold values to judge this quotient. In this way, the third data with the high value in Figure 6 is consequently eliminated. The other corresponding frequency will be saved locally as a true mode.

### 4.3. Identify the Frequency Order

According to the simple string theory, the cable tension is highly related with the linear regression results of ${\left(f_{n}/n\right)}^{2}$ and $n^{2}$. Therefore, the order of all obtained frequencies should be determined correctly. Generally, the continuous true mode frequencies are separated equidistantly in power spectrum density, the frequencies order can be determined through Equation (4).
(4)ni=round(fiα×Δf)
where Δf is the frequency resolution, round is the function to take the nearest integer value. 

With these model frequencies and their physical parameters of the cable, the CTE application can be implemented successfully on an independent sensor node after getting the linear regression results of ${\left(f_{n}/n\right)}^{2}$ and $n^{2}$. 

## 5. CTE Implementation in the Xnode Wireless Sensor Platform

### 5.1. Xnode Wireless Sensor Platform 

The fully automated cable tension estimation algorithm is implemented on a high fidelity smart sensor platform named Xnode as shown in Figure 8, which was developed by Smart Structures and Technology Laboratory at the University of Illinois at Urbana-Champaign [30,31].

A wireless sensor has the advantages of wireless communication, autonomous computation, and of being locally powered with a low cost. Meanwhile, compared with other wireless sensor platforms, Xnode has a higher ADC resolution (24 bits) and a higher sampling rate (16 kHz) which can provide high fidelity measurements for the monitoring research of engineering structures.

The strategy of wireless cable tension monitoring based on Xnode is shown in Figure 8. The entire system consists of one gateway node and several leaf nodes, which communicate with each other via remote command middleware services. The gateway connected to a PC through a USB cable is responsible for managing all leaf nodes by sending task commands and receiving information from leaf nodes. The leaf nodes are mounted on cables to measure their vibration data and process them locally based on applications. 

For a long-span cable-stayed bridge with a large amount of cables, their own properties should be saved in an SD card on the leaf nodes. Furthermore, the wireless sensor can obtain cable parameters from local SD card automatically by executing the task of reading the relevant configuration file. Meanwhile, only the auto power spectrum is used to capture the natural frequencies of the cable through a peaks-picking method and the cable tension then calculated based on the string theory method. During the process, the leaf nodes do not need to share any information between each other. 

To promote the application of CTE in practice, as shown in Figure 9, the sensor nodes should sample acceleration data based initially on sampling parameters which are received from the gateway through the remote command. Once the vibration data are acquired, the Xnode processor of leaf nodes will autonomously process acceleration data locally based on the CTE task. Finally, only the required results will be sent back to the gateway node and shown on PC screen terminal. 

### 5.2. Practical Validation

To verify the wireless cable tension estimation algorithm on Xnode, the data of Jindo Bridge (shown in Figure 10) has been used in this paper. The data of Jindo Bridge were acquired in 2010 in collaboration with the University of Illinois at Urbana-Champaign, KAIST University and the University of Tokyo [32]. The details of the test can be found in S H Sim et al. [25]. 

The data from sensor node 64# (see Figure 11) on the Haenam side were used in Xnode to show the implementation of the cable tension estimation application and demonstrate the effect of cable effective length on the estimation results of cable tension. The sensor node is mounted on cable HC04 which is the shortest one among all monitored cables. The properties of cable HC04 are summarized in Table 1.

The cable tension estimation application is loaded on Xnode to analyze the acceleration data, which is saved in the SD card in advance according to the method mentioned above. The CTE application results are saved in an SD card and also shown on computer screens through the TeraTerm terminal emulator. 

As shown in Figure 12, the power spectral density calculated by Xnode is matched with the results of the Matlab program. Obviously, there are only two peaks picked by using the initial threshold value, which are not enough to estimate cable force correctly. After updating the threshold value several times, the number of found peaks has then been increased to five, and so the correct results are obtained. Meanwhile, the second found peak (a false frequency) is successfully excluded through the method mentioned above. Note that the effective length used to estimate cable tension force is set to be equal to 95.38 m here [33]. 

The results of Xnode displayed on TeraTerm are shown in Figure 13. True frequencies and cable tension force values from Xnode and Matlab are summarized in Table 2. All the frequencies and corresponding orders show excellent agreements. 

As previously mentioned, the peak-picking method result is in accordance with the vibration. Ten arrays of measured data from cable #64 of Jindo Bridge are used to better describe the changes of extracted frequencies and tension force with varied vibrations, and the cable tension force values are listed in Table 3 using the effective length estimated by Park et al. (2008). The RMS value is used to represent vibration magnitude. 

### 5.3. Discussion about Automated Cable Tension Estimation in WSN

The WSN has been widely used in cable-stayed structures to realize long-term health monitoring or reliability assessments. The peak-picking method is the most convenient and efficient way to obtain modal frequencies to estimate cable tension force based on the vibration method. However, it may be difficult to determine the peaks due to noise, sensor sensitivity and some other factors. As for the problems mentioned in Section 3, the automated updating of the threshold value is helpful to capture the potential modal frequencies, especially of weak vibrations with recognizable power spectra. This is despite the way in which the updated threshold value may reduce the reliability of extracted frequencies as the system has corresponding strategies to distinguish undesirable frequencies and determine modal order automatically to ensure its robustness. Compared with previous works in cable tension estimation systems, there are some improvements shown in Table 4. 

The practical validation of the Xnode sensor shows that this system is capable of estimating cable tension with full automation and high robustness. The modal parameters and cable tension greatly agree with the results of Matlab and previous research [25,32]. 

From the results of ten arrays of data, it is found that the peak-picking method failed to acquire frequencies from very weak vibration data, which are instead judged as unrecognizable signals through the method described in Figure 7. Normally as the magnitude of a cable vibration increases, the higher modal frequencies are more easily excited, showing a slight relation with the bigger cable tension force value in Table 3. The small tension force results mainly belong to those data with lower modal frequencies. 

## 6. Conclusions

This paper firstly discussed the existing problems of cable tension estimation using wireless smart sensor networks. To take full advantage of a smart sensor system, a robust and fully automated cable tension estimation method based on ambient vibration was proposed in this study. The reliable estimated frequencies and corresponding mode orders are crucial for cable tension estimation when the simple string theory is adopted. The method in this study is capable of distinguishing the cognizable vibration signal and automatically extract potential frequencies using a peak-picking method. Meanwhile, false frequencies were eliminated successfully and the frequency’s orders were determined correctly to estimate cable tension force. Through this strategy, the fully automated cable tension estimation could be realized for cable-supported structures in long-term health monitoring.

A cable tension estimation application was implemented successfully on the Xnode platform based on the following autonomous processing: calculating and averaging power spectral density, picking peaks from power spectrum, eliminating false peaks, determining mode order and calculating cable tension force. As shown in this case, modal properties were correctly determined without any preprocessing or manual intervention. The natural frequencies and corresponding orders of a Jindo Bridge cable obtained from Xnode system were found to be in excellent agreement with those obtained from Matlab. The estimated tension force results matched very well with about 0.53% difference, it is also close to the results presented by previous studies [25,32] with a difference of less than 0.53%.

The system constructed in this study is effective in automated cable tension estimation. It is sensitive to strong vibration data having obvious peaks in power spectral density. For weak vibration, this system is capable of capturing peaks for accurate tension force by setting an updated threshold value. For invalid vibration data, the results of the cable tension estimation task were set as zero and sent back to the terminal to call for retesting later. 

The Xnode sensor has more memory and an improved processor compared with the previous wireless sensor, which greatly helps to realize cable tension estimation independently, autonomously, and locally. Based on the method proposed in this paper, each cable of a structure would achieve self-identification in long-term monitoring as long as it received the command of a base station node, with only the final results sent back to terminal for early warning.

The method presented in this paper may fail to calculate correct tension force when the power spectrum has continuous and multiple skipped frequencies. Therefore, in real testing, it is better to compare the tension force results of several inspections to obtain a reliable value for structure assessment. Meanwhile, the accuracy of the cable tension force may be affected by the number of obtained frequencies. Generally, the more true frequencies are acquired, the better the accuracy will be. In addition, a recognizable power spectrum which has less obvious peaks may be misjudged as a useless signal; therefore, several tests may be needed to obtain enough peaks in practical experiment. 

The cable tension force of an in-service bridge usually varies in practical monitoring because of vehicle load and environmental effects. Therefore, further research will be focused on improving the algorithm and identification technology, which may also be able to automatically analyze the cause of force changes, especially for sudden changes in long-term monitoring, and send alarms to avoid potential risks for the structures. 

## Figures and Tables

**Figure 1 sensors-21-07229-f001:**
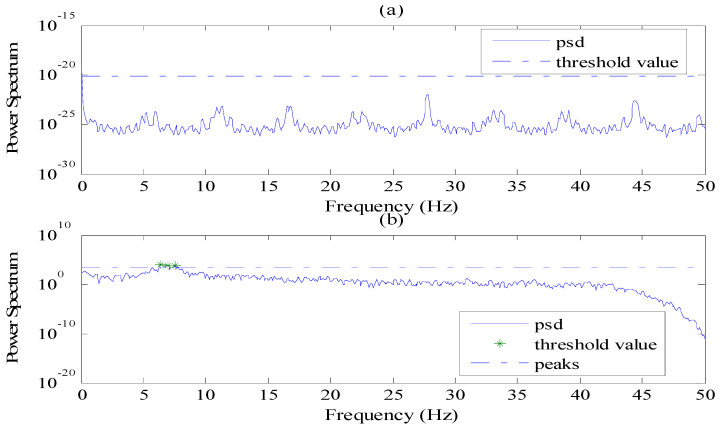
Examples of power spectra that cannot show accurate information about a cable’s frequency. (**a**) A cable vibration that is too weak to exhibit the obvious peaks; (**b**) A power spectral density (PSD) with many false peaks found through the regular peak-picking method.

**Figure 2 sensors-21-07229-f002:**
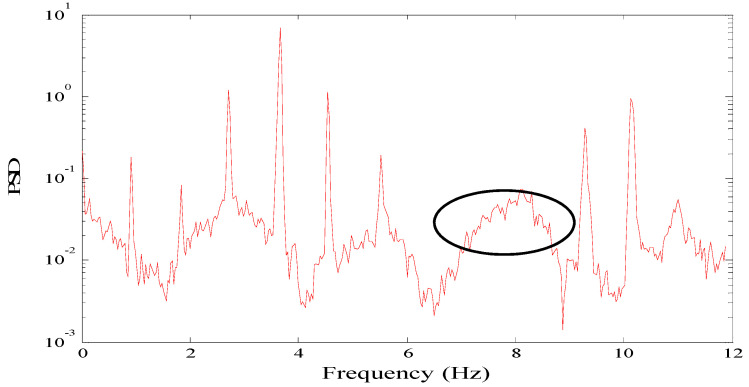
Power spectral density with frequency missing.

**Figure 3 sensors-21-07229-f003:**
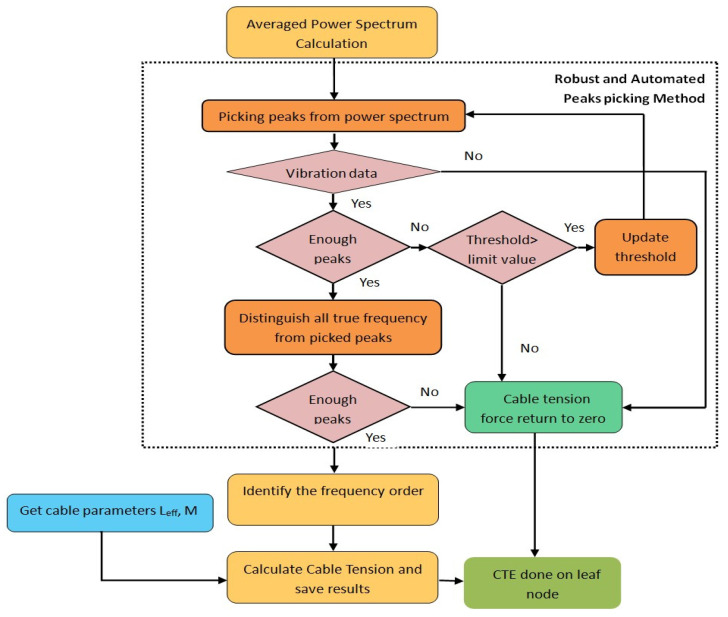
Flowchart of automated and robust cable tension estimation method.

**Figure 4 sensors-21-07229-f004:**
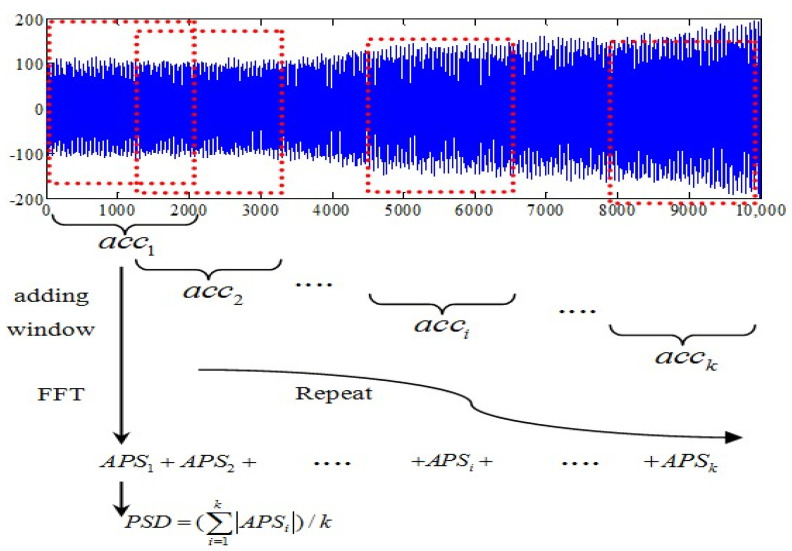
Procedure of fully automated PSD calculation.

**Figure 5 sensors-21-07229-f005:**
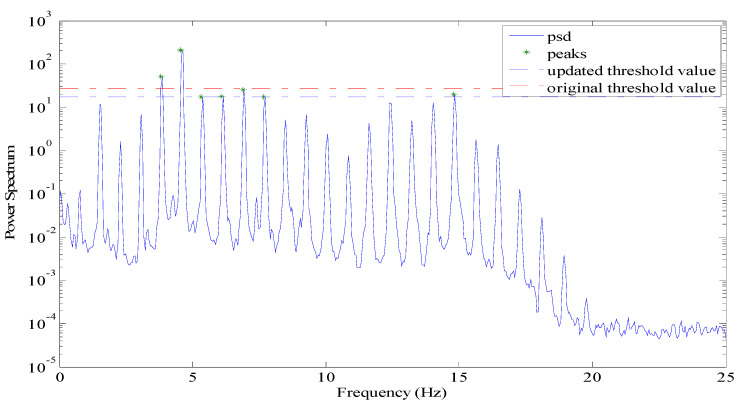
C3 Cable of Jindo Bridge on Haenam side.

**Figure 6 sensors-21-07229-f006:**
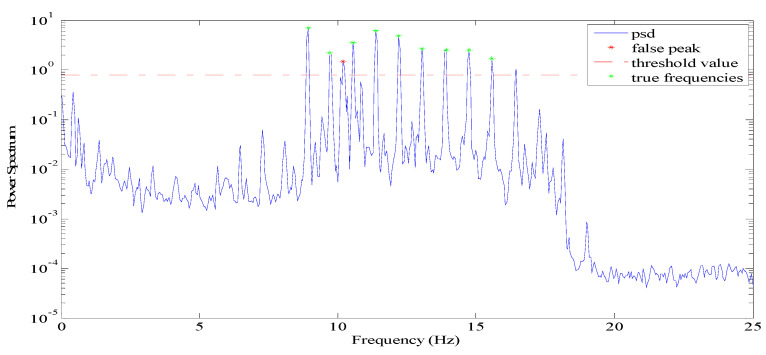
Frequency with false peaks.

**Figure 7 sensors-21-07229-f007:**
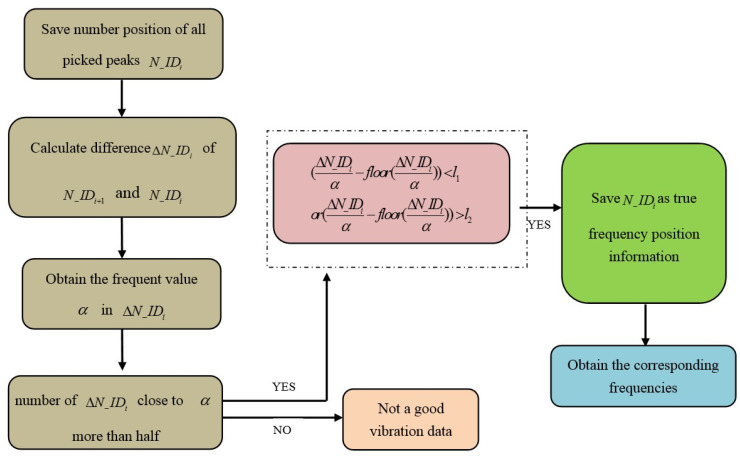
Flowchart of the process for distinguishing true frequencies.

**Figure 8 sensors-21-07229-f008:**
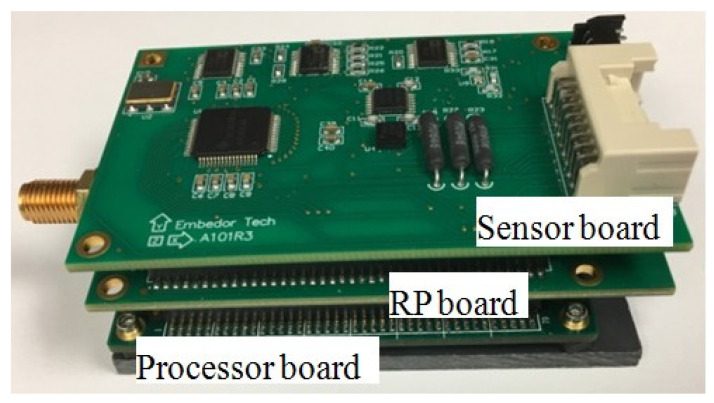
Xnode smart wireless sensor platform.

**Figure 9 sensors-21-07229-f009:**
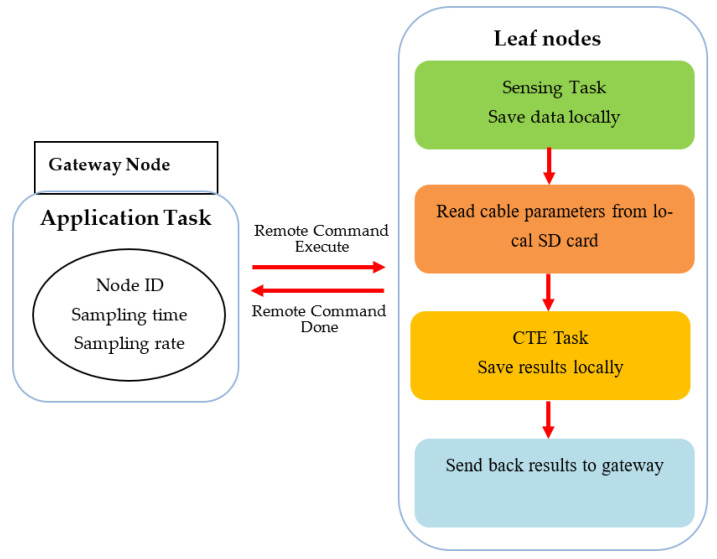
Flowchart of cable tension estimation with Xnode system.

**Figure 10 sensors-21-07229-f010:**
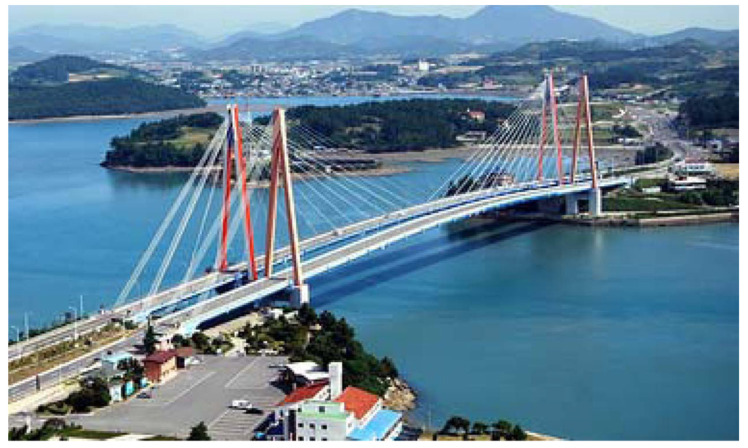
Jindo Bridge.

**Figure 11 sensors-21-07229-f011:**
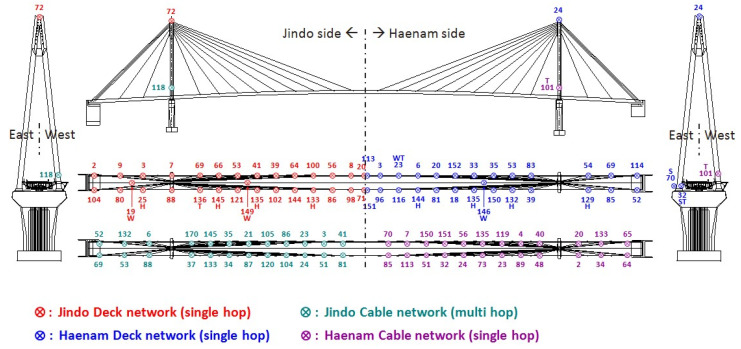
Sensor location.

**Figure 12 sensors-21-07229-f012:**
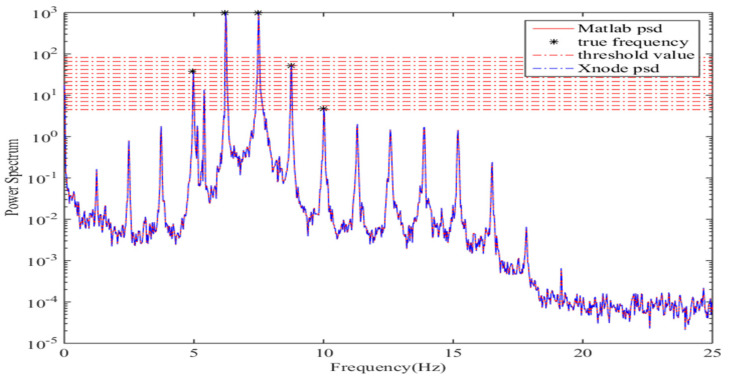
Power spectral density of cable #64 in Z-axis.

**Figure 13 sensors-21-07229-f013:**
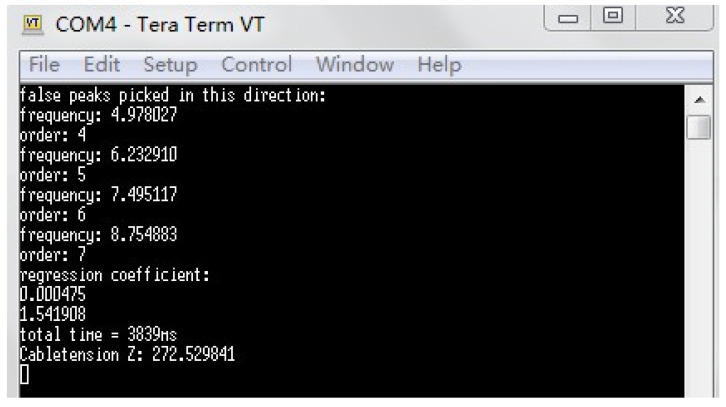
Cable tension estimation results from Xnode.

**Table 1 sensors-21-07229-t001:** Cable properties [33].

Cable		Total Length (m)	Effective Length (m)	Damper Position (m) (from Deck Side)	Net Length (m)	Unit Mass (ton m^−1^)
Sensor No.	Type					
64	ϕ7×151	97.10	95.38	3.40	93.7	0.0476

**Table 2 sensors-21-07229-t002:** Frequency and cable tension force of cable#64.

Frequency Order	Frequency (Hz)	
	Xnode Result	Matlab Result
4	4.9780	4.9710
5	6.2329	6.2193
6	7.4951	7.4863
7	8.7548	8.7530
Tension force (Tonf)	272.53	271.077

**Table 3 sensors-21-07229-t003:** Results of cable tension estimation with varying vibrations.

Data No.		1	2	3	4	5	6	7	8	9	10
**RMS**		3.992	6.943	8.018	8.883	9.908	10.115	10.719	11.541	13.232	32.992
Frequency (Hz)	2nd			2.4813							
	3rd			3.7266							
	4th		4.9706	4.9719		4.9710	4.9709			4.9726	
	5th		6.2167	6.2180	6.2171	6.2193	6.2194	6.2182	6.2200	6.2187	
	6th		7.4836		7.4843	7.4863	7.4854	7.4874	7.4856	7.4849	7.4848
	7th		8.7313		8.7335	8.7530	8.7546	8.7323	8.7344	8.7338	8.7526
	8th		10.0011	10.0244	10.0025			10.0058	10.0020	10.003	10.0007
	9th				11.2711						11.2953
	10th			12.5977							
Regression coefficient			1.5381	1.5369	1.5394	1.5377	1.5333	1.539	1.5404	1.5393	1.5426
Tension force (Tonf)		0 (No peaks)	271.856	271.644	272.086	271.786	271.008	272.015	272.263	272.069	272.652

**Table 4 sensors-21-07229-t004:** Estimation system in this study compared with previous works.

Estimation System	Description
In a previous study [24]	(1) PSD must be recognizable (2) Need the first modal frequency (3) Continuous frequencies (4) Cannot eliminate false frequencies
In a previous study [25]	(1) PSD must be recognizable (2) Continuous frequencies (3) Using constant threshold value (4) Cannot eliminate false frequencies
In a previous study [26,27]	(1) PSD must be recognizable and well separated (2) Picking peaks on phone manually or frequency difference may cause error (3) Not considered robust
In this study	(1) Identify whether PSD is recognizable automatically, if not, return to zero to retest (2) Eliminate false frequencies automatically (3) Update threshold to process small but recognizable vibration data (4) Identify frequency order for both continuous and discontinuous spectra (skipped frequencies)

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
