# Peer review of "Fully Automated and Robust Cable Tension Estimation of Wireless Sensor Networks System"

_sensors, 2021, doi:10.3390/s21217229_

Round 1

Reviewer 1 Report

paper is interesting. covering real investigation on important topic.

results are not well presented but promising.

Unfortunately i recommend to improve related work, references, discussion (missing!), conclusions, but mostly METHODOLOGY!!!

major issue i have with your article coming from definition of the problem you would like to solve in theoretical aspect - and MEthodology you used to solve the theoretical issue == where is main contribution in basic science?

.. it is only vague.. you need to clearly define what is problem, who described problem (use references from major journal), how you would like to solve the problem and comparing other attempt / other solution, your solution/contribution is the best/better/promising..

also limitations of your solution need to be addressed. At conclusions at least..

references need to be places into the text in whole article. number of references need to higher. mostly from journal sources. Q1/Q2 are the most suggested

in the discussion you need to refer to other journal sources and compare your results with other work. and add table where you will summarize your and other contribution to provide more reliable and clear discussion.

also conclusion is too short. It need to include PROS and CONS of your  solution in compare to other solutions or traditional ways. like you have in abstract but more precise. also future directions are missing.

Author Response

The authors appreciate the reviewer’s suggestion.

We have tried our best to revise our manuscript according to the comments, please see the attachment.

Reviewer 2 Report

This paper presents an autonomous cable tension estimation algorithm, which was implemented on the X-node wireless sensor and verified on a real bridge. The topic of the paper is of importance and practical use. The authors provides the detailed introduction of the algorithm, which, in the reviewer's opinion, is viable. However, there are several issues in the paper need to be addressed  before it can be considered for the publication. 

1)  The main focus of this paper is the autonomous cable tension estimation algorithm. However, the discussion in section 2.2 about the effective length of the cable seems not to be complied with the main focus well, which is suggested to be deleted.
2) The words in many pictures in the paper are distorted (e.g., fig 1, 4 etc.). The horizontal or vertical ratio of the pictures need to adjusted.
3) There are two section 2 in the paper.
4)  The title of section 2 can be revised as "Theoretical background of cable tension estimation via cable vibration".
5) The title of section 2 (it should be 3)can be revised as "Key influence factors of cable tension estimation accuracy".
6) The format of many variable symbols in the text need to be changed, which are too large compared with other texts.
7) The English of the paper need to be improved. There are many grammatical errors and inappropriate expressions.

Author Response

(The authors gave the same response as above.)

Round 2

Reviewer 1 Report

Dear Authors,

Thanks for all the corrections. I fine with the extent of the changes you have made to the manuscript. Overall, the quality of the article has increased significantly.

Authors incorporated my suggestion a most in the sense of updated references and deeper description of methods/methodology.

But i still feel, that there is a open space for improvements in highligting pros and cons, updating conclusion and abstract. (based on updated conclusions)  

Thus I suggest to MINOR revisions.

Author Response

(The authors gave the same response as above.)

Reviewer 2 Report

The authors have addressed most of this reviewer's concerns, except for a few minor issues: 

1) The horizontal-vertical ratio of Figure 4 is still not correct.

2) The fonts of the texts in the flowcharts in Figures 7 and 9 are not always the same. It will be better keep uniform.

Author Response

The authors appreciate the reviewer’s suggestion.

We have tried our best to revise our manuscript according to the comments, please see the attachment.

This manuscript is a resubmission of an earlier submission. The following is a list of the peer review reports and author responses from that submission.